# A flow pattern recognition method for gas-liquid two-phase flow based on dilated convolutional channel attention mechanism

Jie Liu⬥, Yang Wu*

School of Intelligent Equipment Engineering, Wuxi Taihu University, Wuxi, China

* wuyl@wxu.edu.cn

## Abstract

Addressing the issue of insufficient key feature extraction leading to low recognition rates in existing deep learning-based flow pattern identification methods, this paper proposes a novel flow pattern image recognition model, Enhanced DenseNet with transfer learning (ED-DenseNet). The model enhances the deep feature extraction capability by introducing a multi-branch structure, incorporating an ECA attention mechanism into Dense Blocks and dilated convolutions into Transition Layers to achieve multi-scale feature extraction and refined channel information processing. Considering the limited scale of the experimental dataset, pretrained DenseNet121 weights on ImageNet were transferred to ED-DenseNet using transfer learning. On a gas-liquid two-phase flow image dataset containing Annular, Bubbly, Churn, Dispersed, and Slug flow patterns, ED-DenseNet achieved an overall recognition accuracy of 97.82%, outperforming state-of-the-art models such as Flow-Hilbert–CNN, especially in complex and transitional flow scenarios. Additionally, the model's generalization and robustness were further validated on a nitrogen condensation two-phase flow dataset, demonstrating superior adaptability compared to other methods.

## 1. Introduction

Gas-liquid two-phase flow plays a critical role in the nuclear industry, including applications in nuclear reactor cooling, nuclear waste processing, gas-cooled reactor technology, and the nuclear fuel cycle [1]. Accurate measurement of two-phase flow parameters is essential for ensuring nuclear safety, improving power generation efficiency, and maintaining the stable operation of equipment. Among these parameters, flow pattern identification is particularly crucial in nuclear thermohydraulic analysis, as the accuracy of heat transfer and fluid mechanics models depends on the precise prediction of flow patterns in pipelines [2]. Therefore, research on gas-liquid two-phase flow pattern identification holds significant scientific and engineering value.

**Data availability statement:** The image dataset and implementation code used in this study are available on GitHub at: https://github.com/liujie556/FlowPattern-EDDenseNet Due to institutional confidentiality policy, the complete dataset and experimental source videos cannot be made fully public. However, representative sample images and code are included in the repository. Additional data may be provided upon reasonable academic request. Institutional Contact for Data Access: Data Access Office, Taihu University Email: researchdata.taihu@gmail.com

**Funding:** This study was partially supported by the Natural Science Research Project of Jiangsu Higher Education Institutions (Grant No. 20KJB210012) and the Qing Lan Project ([2021] No.11). The funders had no role in study design, data collection and analysis, decision to publish, or preparation of the manuscript.

**Competing interests:** The authors have declared that no competing interests exist.

Current approaches for gas-liquid two-phase flow pattern identification can be categorized into manual classification, signal processing, and process tomography. Manual classification relies on experimental personnel visually observing the distribution of two-phase flow within transparent pipes [3]. However, this method is highly subjective and error-prone, particularly in high-speed flows where the phase interface is difficult to distinguish. Signal processing-based methods analyze sensor signals to infer flow patterns [4–6]. Although time-frequency analysis techniques can extract key features such as mean values and frequency distributions, the selection of these features often involves subjective choices, which may lead to inconsistent classification results. Process tomography reconstructs the three-dimensional distribution of gas-liquid flow within pipes [7,8]. This method is non-invasive and provides visualization, making it widely used in flow pattern identification. However, it requires a large number of sensors and computationally expensive image reconstruction, limiting its real-time applicability.

With the rapid development of artificial intelligence, machine learning methods have emerged as promising alternatives for two-phase flow pattern classification [9–11]. Convolutional neural networks (CNNs), known for their strong nonlinear feature extraction capabilities, have revolutionized flow pattern identification. Yang et al. [12] were the first to apply CNNs for flow pattern classification in small horizontal pipes, achieving an average accuracy of 96.2%. Du et al. [13] explored various CNN architectures (LeNet5, AlexNet, and VGGNet16) for oil-water two-phase flow, with classification accuracies ranging from 57.7% to 99.4%. Yang et al. [14] combined electrical capacitance tomography (ECT) with neural networks, leveraging a multi-task learning mechanism, resulting in an accuracy exceeding 98%. However, tomography-based approaches are limited by high hardware costs and data latency, making real-time implementation difficult. Su et al. [15] employed ultrasonic reflection signals to extract statistical features for oil-gas-water Slug Flow classification, using an RBF neural network, achieving a 95.7% accuracy. However, manual feature selection remains a limitation.

Despite these advancements, existing CNN-based methods still face challenges in handling complex flow structures and dynamic changes. Recent state-of-the-art (SOTA) approaches have attempted to address these issues. Lv et al. [16] proposed an optical flow algorithm for small-channel gas-liquid flow pattern recognition, demonstrating improved accuracy in detecting transient flow characteristics. However, this method relies on motion-based analysis using multi-frame sequences, whereas our study focuses on static image classification of gas-liquid flow patterns. Due to fundamental differences in methodology and input data structure, a direct comparison with Lv et al. [16] is not feasible. Zhang et al. [17] introduced a Flow-Hilbert–CNN hybrid model, integrating Flow-Hilbert transformation for signal processing with a CNN-based classifier to improve recognition performance. While this method effectively enhances feature extraction for complex flow structures, it incurs significant computational overhead due to Hilbert transform operations. Additionally, it struggles with the recognition of flow patterns with highly irregular gas-liquid distributions, where dynamic changes in phase boundaries are not well captured by frequency-domain features. Given its CNN-based architecture, this method serves as a relevant

benchmark for comparison with our approach. Nevertheless, existing CNN-based methods have three major limitations. First, most CNN models prioritize spatial convolutional features but fail to enhance the most relevant channels, reducing classification efficiency. Second, conventional CNNs typically rely on fixed-size convolutional kernels, which may fail to capture flow structures at different scales, leading to reduced generalization in dynamic flow conditions. Third, many studies attempt to improve classification accuracy by increasing the depth or width of CNN architectures, which significantly increases computational complexity and memory requirements.

To address these limitations, this study proposes an enhanced DenseNet-based model (ED-DenseNet) that integrates two key improvements. First, the Efficient Channel Attention (ECA) Mechanism enhances channel-wise feature representation, allowing the network to selectively emphasize the most relevant flow pattern features. Second, the use of dilated convolutional layers with varying dilation rates allows the proposed model to capture multi-scale feature dependencies without increasing network depth or computational overhead. Additionally, to further assess the model's robustness, we introduce a transitional flow test dataset consisting of flow patterns at intermediate states (e.g., Slug→Churn, Churn→Annular), which better reflect real-world industrial conditions.

The key contributions of this study are summarized as follows:

1. For the first time, the integration of the ECA attention mechanism and dilated convolutions is applied to static image-based two-phase flow pattern recognition, enhancing multi-scale feature extraction and channel information representation capabilities.

2. A redesigned multi-branch DenseNet architecture is developed to strengthen deep feature learning and improve classification accuracy.

3. Transfer learning techniques are introduced to alleviate overfitting on small-scale datasets, significantly accelerating model convergence and enhancing generalization performance.

4. A test dataset containing ambiguous and transitional flow patterns is constructed to comprehensively evaluate the model's robustness and adaptability under complex and dynamic flow conditions.

Unlike previous studies, our approach effectively enhances feature representation through attention mechanisms and expands receptive fields via dilated convolutions. As a result, ED-DenseNet achieves state-of-the-art accuracy in gas-liquid two-phase flow classification without significantly increasing computational complexity. Extensive experiments validate the effectiveness of this approach, showing robust performance across various datasets and challenging flow conditions.

## 2. Materials and methods

### 2.1 Flow pattern image acquisition

Flow patterns refer to the different states or modes exhibited by fluids' flow characteristics and morphology in gas-liquid two-phase flow [18]. Table 1 describes the types and characteristics of two-phase flow patterns in vertical upward flow. Different gas-liquid two-phase flow patterns have varying effects and influences on nuclear industry production. For instance, Bubbly Flow can lead to the formation of bubbles in the coolant within a nuclear reactor, causing local turbulence in the coolant and affecting heat transfer and the cooling efficiency of the reactor [19]. Similarly, Slug Flow can cause local variations in coolant velocity within cooling pipes, impacting heat transfer and temperature distribution, thereby reducing the efficiency and stability of reactors or processing equipment [20]. Therefore, analyzing and controlling two-phase flow patterns is crucial to ensure safe, stable operation and performance optimization in the nuclear industry. To obtain a dataset of gas-liquid two-phase flow images for different flow patterns, which is necessary for training convolutional neural network models, flow pattern image acquisition experiments were conducted on a gas-liquid two-phase flow experimental system. Fig 1 illustrates the schematic of the gas-liquid two-phase flow experimental system.

**Table 1. Flow pattern types and characteristics.**

| Flow Patterns | Main Characteristics |
|---|---|
| Bubbly Flow | Gas is dispersed in the liquid through tiny bubbles, presenting a discrete gas phase. The liquid phase fills the spaces between the bubbles to form a continuous phase, with the gas phase exhibiting discontinuity. |
| Slug Flow | Developed from Bubbly Flow, as the gas flow rate increases, the bubbles in the liquid gradually grow and then coalesce into larger bubble shapes. |
| Churn Flow | With further increase in gas flow rate, the bubbles in Slug Flow collide, break, and deform, mixing with the liquid to form a turbulent mixture that churns up and down. |
| Annular Flow | As the gas volume increases, the liquid phase fills the entire pipe wall as a liquid film, while the main flow through the centre is gas, with droplets still present. In this flow pattern, the gas and liquid phases become continuous again. |
| Dispersed Flow | The flow velocities of each phase primarily determine it, as the interface between phases continuously changes. With increasing flow velocity, Dispersed Flow undergoes coalescence and eventually transitions into a continuous phase, while the original continuous phase may transform into a Dispersed Flow. |

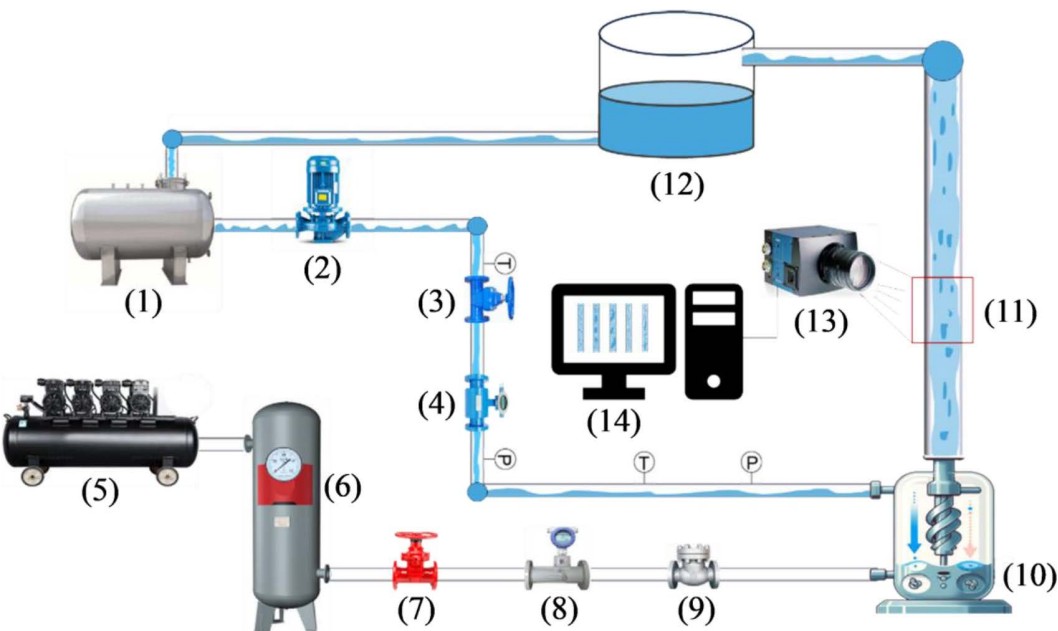

**Fig 1. Schematic diagram of the experimental setup:** (1) Water tank, (2) Centrifugal pump, (3) Valve, (4) Turbine flowmeter, (5) Air compressor, (6) Air buffer tank, (7) Needle valve, (8) Gas flowmeter, (9) Check valve, (10) Gas-liquid mixer, (11) Observation section, (12) Gas-liquid separation water tank, (13) High-speed camera, (14) Computer.

In the experiment, air and water were used as the working media. Air was supplied by an air compressor, passed through a gas buffer tank to reduce pressure fluctuations, and its flow rate was precisely controlled by a gas flowmeter. Water from a tank was pumped out by a centrifugal pump and delivered to the experimental pipeline. The flow rate was adjusted by regulating the valve opening, and the flow was measured by a turbine flowmeter. To prevent water from the gas-liquid mixer from flowing back into the gas flowmeter, a check valve was installed after the gas flowmeter. Air and water were mixed in a gas-liquid mixer at the bottom of the vertical tube, and the mixed two-phase flow moved upward through a circular tube with an inner diameter of 20 mm. The flow pattern observation section was located 80 tube diameters away from the mixer outlet. The long distance was set to ensure fully developed flow [21,22], allowing the two-phase fluid to conform to the "solidification" flow hypothesis [23] when passing through the flow pattern observation section. The

flow pattern observation section was made of high-transparency quartz glass, allowing direct observation of the fluid flow inside the tube. Flow pattern images were captured using a high-speed camera, the EoSens@CUBE6, produced by the German company MIKROTRON. This high-speed camera can shoot at up to 112,000 frames per second with a resolution of 1280×1024 pixels. For this experiment, it was set to 2000 frames per second. After shooting, the fluid flowed out of the tube. The exiting two-phase flow was separated by gravity, with air being released into the atmosphere and water flowing back into a gas-liquid separation reservoir under gravity, returning to the water tank via piping.

Upon determining the overall system structure and the properties of the two-phase fluid, the flow pattern of the gas-liquid two-phase flow primarily depends on the flow rates of the gas and liquid phases. By adjusting the valve openings in the liquid pipeline in conjunction with a turbine flowmeter to control the liquid phase flow rate, and by adjusting the valve openings in the gas pipeline in conjunction with a gas flowmeter to control the gas phase flow rate, different flow patterns can be generated. During the experiment, the pressure in the flow pattern observation section was maintained between 0.1 and 0.35 MPa, the temperature range of the two-phase medium was 21–24 °C, the liquid phase volumetric flow rate ranged from 0.003 to 1.73 m³/h, and the gas phase volumetric flow rate ranged from 0.02 to 20.98 m³/h. The observed flow patterns included Annular Flow, Bubbly Flow, Churn Flow, Dispersed Flow, and Slug Flow.

## 2.2 Dataset production

**2.2.1 Image preprocessing.** Images captured directly from the flow pattern observation section are affected by imaging technology, hardware limitations, lighting, residue impurities in the pipeline, and vibrations present during the experiment, which introduce noise into the gas-liquid two-phase flow images. Therefore, it is necessary to preprocess the images to reduce the impact of noise on the subsequent analysis process. By using median filtering and contrast stretching techniques for preprocessing, the primary objectives are to eliminate salt-and-pepper noise in the images and to adjust the grayscale distribution to enhance contrast, thereby simplifying the data, increasing the detectability of crucial information and improving the accuracy of flow pattern feature extraction.

The results of the preprocessed images of the five types of vertical annular gas-liquid two-phase flow patterns obtained in the experiment are shown in Fig 2. It can be observed from the figure that some irrelevant noise has been eliminated, and both the contrast and quality of the images have been improved.

**2.2.2 Dataset partitioning.** In the experiment, images captured by a high-speed camera were extracted at fixed frame intervals to obtain an initial dataset of gas-liquid two-phase flow images. The image dataset was then screened, and images with apparent flow pattern features and high clarity were selected for preprocessing. Subsequently, manual classification and annotation were performed. Due to the visual difficulty in identifying transitional scenarios between different flow patterns, the overlapping nature of flow pattern features, and the subjectivity of manual classification [24], there has yet to be a consensus on the number of gas-liquid two-phase flow patterns. In our custom dataset, we categorized the flow pattern images based on the feature descriptions provided in Table 1, resulting in five typical flow patterns: Annular Flow, Bubbly Flow, Churn Flow, Dispersed Flow, and Slug Flow, as shown in Fig 2.

After screening and preprocessing, 18,000 flow pattern image samples were obtained, with 3,600 samples for each flow pattern, serving as the gas-liquid two-phase flow image dataset required for the experiment. The image dataset was divided into a training set (70%), a validation set (20%), and a test set (10%) using a non-replacement random sampling method. Due to the size of the gas-liquid two-phase flow image dataset, data augmentation was performed on the training data to enhance the model's generalization. Random horizontal and vertical flips were applied to each mini-batch of images read using PyTorch functions. By randomly flipping the images, the model's features of interest can appear in different positions within the images, reducing the model's dependence on the location of feature occurrence and thereby improving the model's generalization capability.

In addition, in order to further evaluate the robustness of the model in practical applications, this study also constructed an additional transition flow test set, and the typical image is shown in Fig 3. It is specifically used to evaluate the

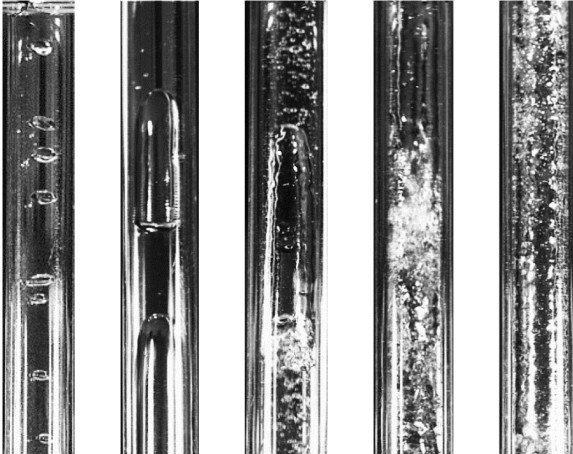

**Fig 2. Results after preprocessing.**

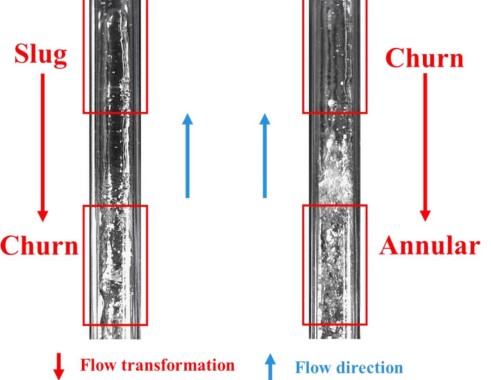

**Fig 3. Transitional flow pattern images.**

classification performance of the model when the flow boundary is fuzzy. This dataset particularly focuses on two typical transitional flow processes:

- Slug to Churn transition: In this stage, Taylor bubbles gradually break up, liquid-phase turbulence intensifies, and the flow state becomes highly unstable.

- Churn to Annular transition: During this transition, the liquid film gradually forms along the pipe walls, while the gas phase increasingly concentrates at the center of the pipe, but has not yet fully developed into a stable annular flow pattern.

To build this dataset, 1,000 transitional flow images were carefully selected from experimental video frames, including transitional samples between slug and churn flows, as well as between churn and annular flows. These transitional flow samples exhibit characteristics of both adjacent flow patterns, making them inherently ambiguous. The Transitional Flow Test Set is not included in the model training process and is exclusively used for evaluation. It provides a challenging classification scenario to test the model's ability to recognize complex flow transitions.

This enhancement ensures that the dataset better reflects real industrial conditions, where gas-liquid two-phase flow patterns often have gradual rather than sharply defined boundaries. By incorporating the Transitional Flow Test Set, this study enables a more comprehensive assessment of the model's applicability in real-world environments, ensuring that it can not only accurately classify well-defined flow patterns but also maintain high recognition accuracy in complex and uncertain flow conditions.

### 2.3 Flow pattern recognition model construction

This paper adopts the DenseNet121 [25] model as the baseline network. However, the original Dense Block only considers the correlation between different layers, neglecting the extraction of channel information during the image recognition process, which to some extent affects the accuracy of the model for flow pattern recognition tasks. The ECA attention mechanism and dilated convolution module are introduced to enable the model to acquire channel information and enhance its feature extraction capabilities. Utilizing these two modules to optimize the DenseNet121 network, a flow pattern recognition method based on attention mechanisms and DenseNet121, termed ED-DenseNet (ECA-DC-DenseNet), is proposed.

**2.3.1 ECA attention mechanism.** An attention mechanism is incorporated into the network to address the issue in flow pattern recognition tasks where the task is susceptible to interference from the quartz glass tube walls on both sides and irrelevant background. The attention mechanism can locate the information of interest by training weights on the channel dimension of the feature layers extracted by the backbone network, allowing the network to focus on the gas-liquid phase part of the input images, thereby suppressing useless information from the surrounding complex environment and mitigating the impact of the complex environment on the flow pattern recognition task, leading to an improvement in the performance of the deep convolutional network. The network's backbone is a critical feature extraction component; therefore, the ECA attention mechanism [26] is added to the Dense Block portion of the DenseNet network.

The channel attention mechanism has been proven to have great potential in enhancing the performance of deep convolutional neural networks, as it can focus on local information in images and suppress irrelevant information. The channel attention module (Squeeze and Excitation, SE) in the Squeeze and Excitation Network (SENet) captures nonlinear cross-channel interactions through Global Average Pooling (GAP) and two Fully Connected (FC) layers and finally uses the Sigmoid function to obtain weights for different channels. The FC layers involve dimensionality reduction to control model complexity, which, although widely used in subsequent channel attention modules, can be avoided to learn more effective channel attention by analyzing the impact of dimensionality reduction and cross-channel interaction in the SE module. Therefore, the ECA attention module considers using one-dimensional convolutions instead of FC layers to capture local cross-channel interactions through each channel and its adjacent $K$ channels, which can improve classification accuracy more effectively. The ECA structure is shown in Fig 4.

**2.3.2 Dilated convolution.** To enhance the model's feature extraction capabilities, dilated convolutions [27] are introduced. Unlike ordinary convolutions, dilated convolutions inject gaps into the standard convolutional kernels, which not only increases the model's receptive field but also captures multi-scale contextual information. Dilated convolutions were initially proposed to address the problem of image segmentation. Standard image segmentation algorithms typically use pooling layers and convolutional layers to increase the receptive field, which also reduces the resolution of the feature maps. The feature maps are then enlarged through upsampling to restore the image size. The process of shrinking and enlarging the feature maps results in a loss of precision. Therefore, an operation is needed that can increase the receptive field while keeping the size of the feature maps unchanged, thus replacing the downsampling and upsampling operations. Based on the above requirements, dilated convolutions emerged.

Dilated convolution refers to expanding the convolutional kernel by inserting zeros between the elements of the kernel, as specifically illustrated in Fig 5. If $a$ represents the dilation rate and $k$ is the original size of the convolutional kernel, then after the introduction of dilated convolution, the size becomes $k = k + (k-1)(a-1)$. For example, the receptive fields

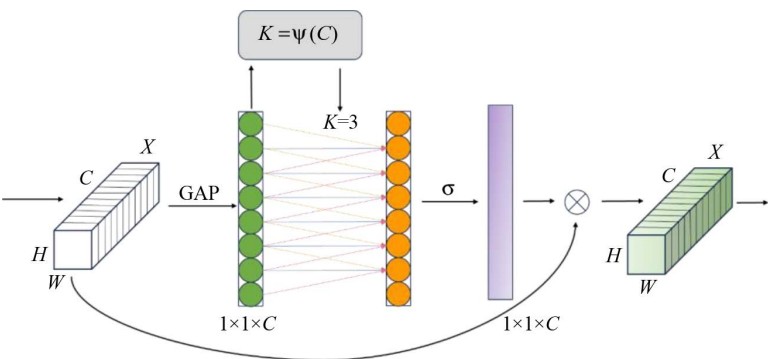

**Fig 4. ECA module structure.**

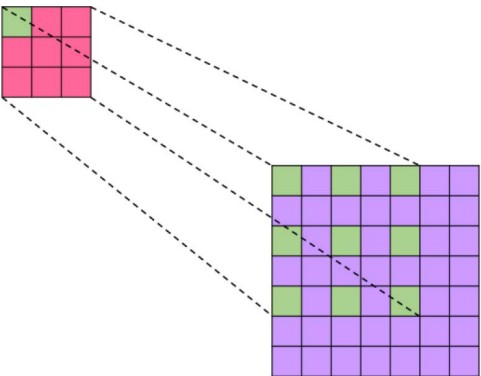

**Fig 5. Working principle of dilated convolution.**

for $a = 1, 2, 4$ are shown in Fig 6. It can be observed from Fig 5 that dilated convolution can increase the receptive field of the output unit without enlarging the size of the convolutional kernel. When convolutions with multiple dilation rates are stacked, the different receptive fields can bring multi-scale information, capturing contextual information at multiple scales. This paper introduces dilated convolution into the Dense Block network structure and the Transition Layer, acquiring equivalent feature maps under different receptive fields through dilated convolutions with dilation rates of 1 and 3, respectively.

**2.3.3 Flow pattern recognition model based on ED-DenseNet.** DenseNet addresses the degradation problem in developing deep neural networks by establishing dense connections between layers, using the feature maps from all preceding layers as inputs to the current layer. This approach significantly enhances feature propagation and reuse. Furthermore, by eliminating the need to relearn redundant feature mappings, DenseNet reduces the number of parameters in the network. These characteristics enable it to perform better with less computational resources, making it suitable for industrial applications. The structure of the model is illustrated in Fig 7.

The overall model consists of four Dense Blocks, interconnected by Transition Layers. As its core module, the structure of a Dense Block is illustrated in Fig 8. The essential components that constitute a Dense Block typically include 1×1 and 3×3 convolutions and dense connections. The role of the 1×1 convolution operation is to reduce dimensions by decreasing the number of feature map channels, thereby reducing computational load while retaining essential features. The 3×3 convolution operation extracts higher-level features from the input feature maps. The network can gradually learn more

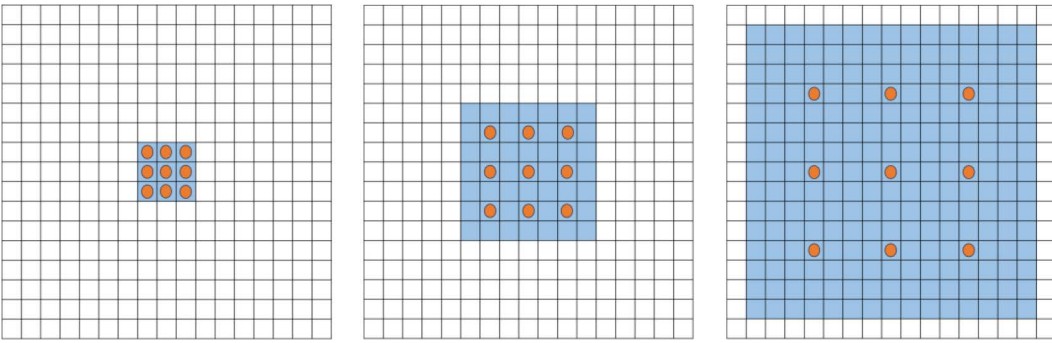

**Fig 6. Receptive fields under different dilation rates.**

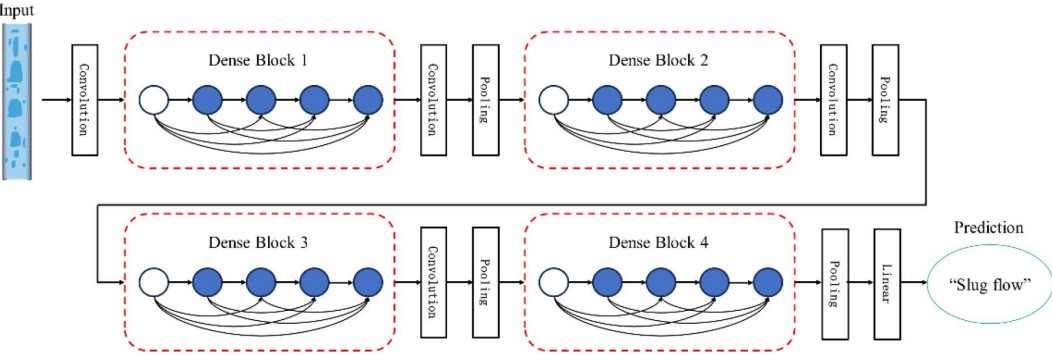

**Fig 7. DenseNet network architecture diagram.**

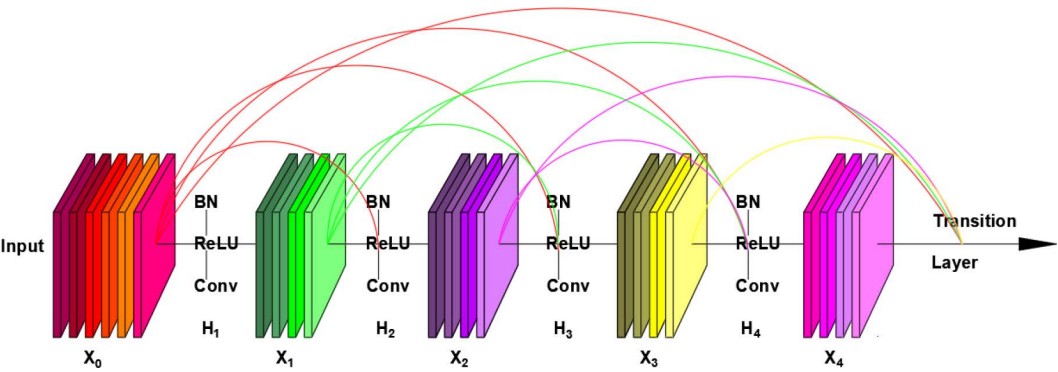

**Fig 8. Dense Block structure diagram.**

abstract and complex features by stacking multiple such convolutional layers. This structure, alternating between 1 × 1 and 3 × 3 convolutions, is a critical component of the Dense Block. It not only lowers the dimensionality of features but also increases the nonlinearity of the network, preventing model overfitting. In the four Dense Block modules of DenseNet121, the number of stacked pairs of 1 × 1 and 3 × 3 convolutions are 6, 12, 24, and 16, respectively. The specific composition of these stacks is as follows:

Consider a feature layer $x_0$ passed through a network model with $L$ layers. In traditional convolutional neural networks, the nonlinear transformation of the $i$-th layer is denoted as $H_i(*)$, and the output of the feature layer of the $i$-th layer is denoted as $x_i$. The relationship can be expressed by the following formula:

$$x_i = H_i(x_0) \tag{1}$$

To further optimize the efficiency of information utilization, DenseNet introduces the Dense Block model, which correlates the input of the $i$-th layer with the outputs of all preceding layers in an $L$-layer network model. This is represented by the formula:

$$x_i = H_i([x_0, x_1, \ldots, x_{i-1}]) \tag{2}$$

In this context, $[x_0, x_1, \ldots, x_{i-1}]$ represents the concatenation operation of feature maps, where the outputs of all preceding feature layers are stacked together along the channel dimension. The nonlinear transformation $H_i$ refers to the structure composed of alternating 1 × 1 and 3 × 3 convolutions as mentioned earlier. Due to the feature vector connections between every convolutional structure within the Dense Block, the information density and the flow of gradient propagation throughout the network are improved, allowing the model to achieve better feature utilization efficiency. During the calculation of gradient backpropagation, each layer of the network can receive gradient information passed from subsequent feature layers, which helps to mitigate the vanishing gradient problem that often arises with increased network depth. Furthermore, the more densely connected structures typically exhibit a regularization effect in deeper model architectures, effectively preventing the model from overfitting during training.

The Transition Layer, which serves as a connecting layer, incorporates a 1 × 1 convolution and a 2 × 2 average pooling operation. This pooling operation is utilized to reduce the spatial dimensions of the feature maps, effectively decreasing their length and width, thereby lowering the spatial complexity of the model. The role of the Transition Layer is to perform dimensional adjustments and compress the size of the feature maps between Dense Blocks, facilitating better control of the information flow, reducing the number of parameters, and promoting efficient training and learning of the network. This helps to ensure that the network can more effectively extract and propagate useful features in subsequent layers.

Based on the ECA and dilated convolution structures described in Sections 1.3.1 and 1.3.2, this paper presents a modification to the DenseNet121 architecture. DenseNet is a deep convolutional neural network architecture composed of Dense Blocks. DenseNet-121 is one variant of this architecture with a depth of 121 layers, comprising four Dense Blocks. Each Dense Block internally contains a number of Dense Layers.

In the original Dense Layer, two 1 × 1 convolutions and a 3 × 3 convolution are connected in series to reduce dimensions and extract features. To enhance its feature extraction capabilities and introduce channel attention, this paper proposes an improvement to the Dense Layer. The modified Dense Layer retains the original 1 × 1 and 3 × 3 convolution sizes but incorporates dilation rates of 1 and 3 into their respective convolutional kernels. Compared to standard convolutional kernels, using dilated convolutions allows for the extraction of multi-scale features through an enlarged receptive field without additional computational cost. Unlike the previous serial structure, the modified structure, as shown in Fig 9, consists of three branches formed by convolutions of different sizes, each learning features from the input information. Different types of convolutions capture information at different scales, and the features obtained from the three branches are then added together under a parallel structure to enhance the model's learning ability. Subsequently, the ECA attention mechanism is employed to strengthen the channel attention of the input information, clearly classifying pixels. Finally, Dropout and the ReLU function are used to optimize the training parameters.

In the original Transition Layer, a 1 × 1 convolutional layer and a 2 × 2 average pooling layer are used to control the feature maps' size and number of channels. To further extract multi-scale information while retaining the 1 × 1 convolution, an additional 5 × 5 dilated convolutional module is introduced, and the ReLU activation function is replaced with

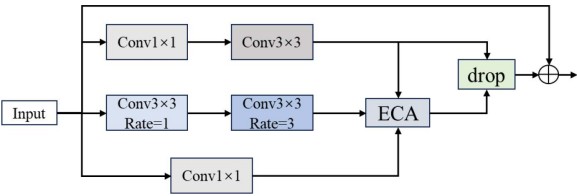

**Fig 9. Dense Layer structure diagram.**

the GELU activation function. Compared to the ReLU activation function, the GELU activation function is smoother, which helps to improve the convergence speed and performance of the training process and reduce computational costs. Finally, the extracted flow pattern features undergo Batch Normalization (BN), Global Average Pooling (GAP), and Fully Connected (FC) operations within the classification network. The probabilities for each flow pattern in the flow pattern image are calculated using the SoftMax function, and the category with the maximum value is output as the classification result, achieving flow pattern recognition. The proposed ED-DenseNet network structure is illustrated in Fig 10.

### 2.4 Optimizer algorithm and model evaluation methods

**2.4.1 Optimizer algorithm.** In deep learning, the choice of optimization algorithm determines how model parameters are updated, directly influencing convergence speed and final performance. The original Stochastic Gradient Descent (SGD) algorithm [28] used in DenseNet121 enables rapid convergence; however, selecting an appropriate learning rate poses a challenge. A small learning rate results in slow convergence, while a large learning rate can lead to significant fluctuations, making the model prone to settling in local optima or saddle points.

To accelerate the convergence of model training, this study employs the Adam optimization algorithm [29] to replace the original SGD. Adam integrates the advantages of Adaptive Gradient Algorithm (AdaGrad) and Root Mean Square Propagation (RMSprop), effectively preventing vanishing gradients in the early training stages while ensuring stable training and faster convergence. The parameter update process in Adam is given by:

$$m_t = \beta_1 \cdot m_{t-1} + (1 - \beta_1) \cdot g_t$$

$$v_t = \beta_2 \cdot v_{t-1} + (1 - \beta_2) \cdot g_t^2$$

$$\theta_t = \theta_{t-1} - \frac{\alpha}{\sqrt{v_t} + \epsilon} \cdot m_t$$

(3)

where $m_t$ and $v_t$ represent the first and second moment estimates of the gradients, $\alpha$ is the learning rate, and $\epsilon$ is a small constant to prevent division by zero. Compared to SGD, Adam dynamically adjusts the learning rate for each parameter, making updates more stable and reducing the need for extensive hyperparameter tuning.

However, Adam may lack robustness when dealing with non-stationary objective functions or abrupt gradient changes. To address this issue, this study further explores AdaMax [30], a variant of Adam that uses the infinity norm (max-norm) for gradient updates. This modification enhances the optimizer's robustness to gradient variations, improving model stability. Finally, we compare Adam and AdaMax in optimizing the ED-DenseNet model and related architectures to determine the most effective optimization strategy.

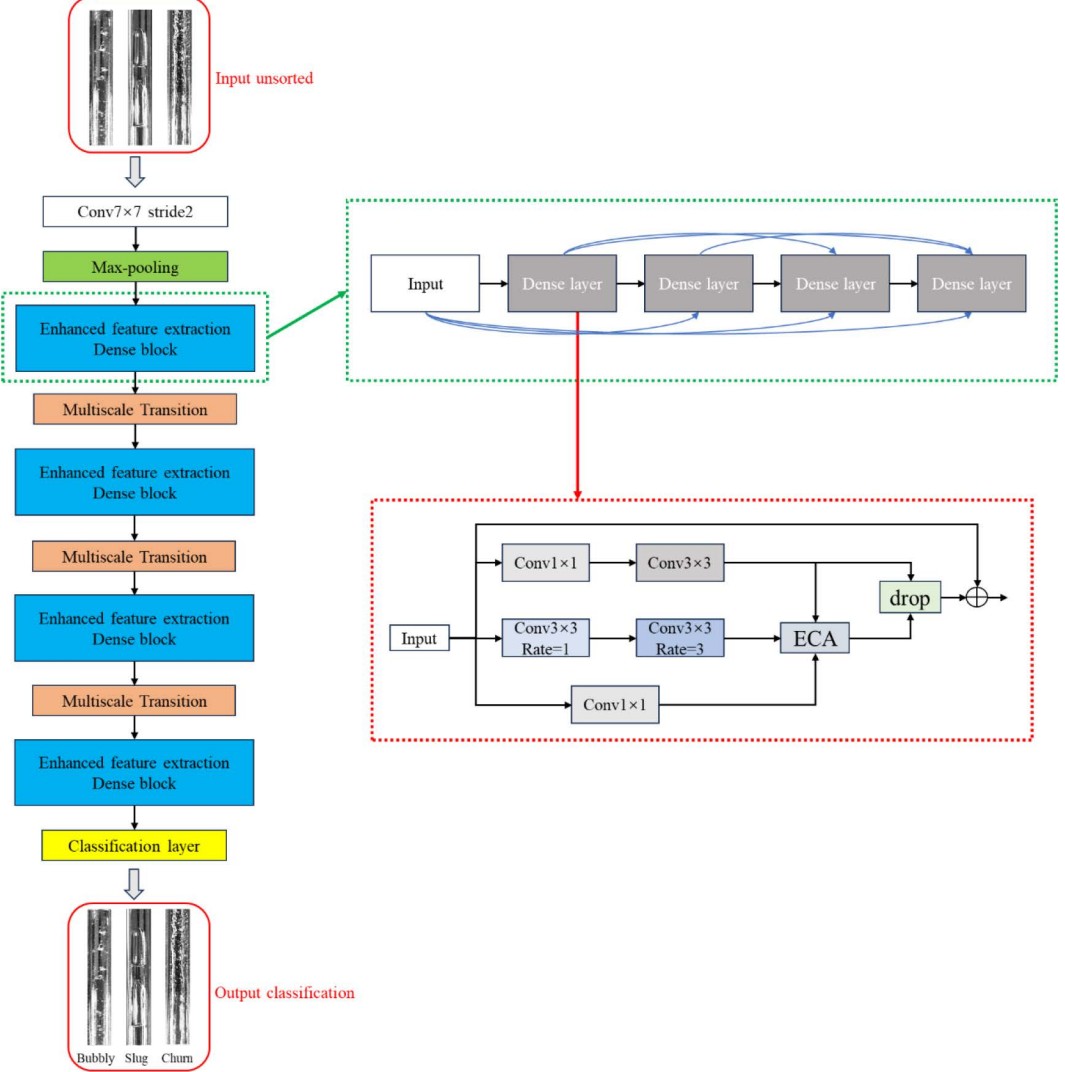

**Fig 10. Schematic diagram of ED-DenseNet model structure.**

**2.4.2 Model evaluation methods.** The classification performance of the model in gas-liquid two-phase flow pattern recognition is assessed using the following key metrics [31]:

Accuracy:

$$Accuracy = \frac{TP + TN}{TP + TN + FP + FN} \tag{4}$$

Measures the proportion of correctly classified samples relative to the total number of samples.

Recall:

$$Recall = \frac{TP}{TP + FN} \tag{5}$$

Evaluates the model's ability to correctly identify positive class (target) samples.
   Precision:

$$Precision = \frac{TP}{TP + FP} \qquad (6)$$

Assesses the proportion of correctly predicted positive samples among all predicted positives.
   F1-score:

$$F1Score = \frac{2 \times Precision \times Recall}{Precision + Recall} \qquad (7)$$

A harmonic mean of Precision and Recall, particularly useful in imbalanced datasets.
   Additionally, to comprehensively evaluate the model's practical applicability, we also assess: Computational efficiency (training time, inference time, and memory usage), Robustness (performance stability under varying flow conditions) and Generalization ability (model adaptability across different datasets).
   By incorporating these metrics, we systematically analyze the performance of ED-DenseNet and compare it against state-of-the-art (SOTA) methods to ensure fair evaluation.

## 2.5 Transfer learning

Transfer learning [32] involves leveraging the commonalities between different learning tasks to transfer knowledge across tasks, enabling the application of knowledge learned from existing data or environments to new data or environments. The dataset distribution in this paper is relatively concentrated, with fewer categories and fine-tuning models pre-trained on high-quality datasets through transfer learning can effectively address issues such as overfitting. Therefore, this paper retains the weight parameters of the DensNet121 network that have been trained on the original training dataset. It transfers them to the constructed ED-DenseNet model, allowing the model to acquire relatively rich prior knowledge. On this basis, the flow pattern dataset is used for learning and training. The original training dataset is the ImageNet classification dataset, which has 1000 categories, including 1.2 million training images, 50,000 validation images, and 100,000 test images.
   Fig 11 illustrates the structure of the transfer network model used for flow pattern recognition. The parameters of the pre-trained network are used as the initial parameters for the model being constructed to complete the weight initialization. Subsequently, the ED-DenseNet model is modified by changing the last fully connected layer to have five outputs, making the model suitable for the 5-class classification problem corresponding to the five types of gas-liquid two-phase flow patterns. Additionally, the SoftMax function is selected as the activation function for the last layer, and the loss function is chosen as the categorical cross-entropy.

## 3. Results and analysis

The experimental platform is equipped with an Intel i9-12900K processor, 64GB of memory, and an NVIDIA RTX A4000 graphics card. The software environment includes the Windows 10 operating system and utilizes an Anaconda virtual environment. The programming language is Python 3.9, and the model is implemented using the PyTorch deep learning framework with GPU acceleration. Data analysis is conducted using the Origin 2017 version.
   During the experiment, all models were initially trained using the Adam optimizer with the exponential decay rate for the first-moment estimates set to 0.9, the exponential decay rate for the second-moment estimates set to 0.999, and the initial learning rate set to 0.01. The input image size for the models was 224×224 pixels. Due to hardware constraints, the batch size was set to 32, and the number of epochs was set to 50. The models were trained in a mini-batch manner to enhance the generalization capabilities of the deep learning models.

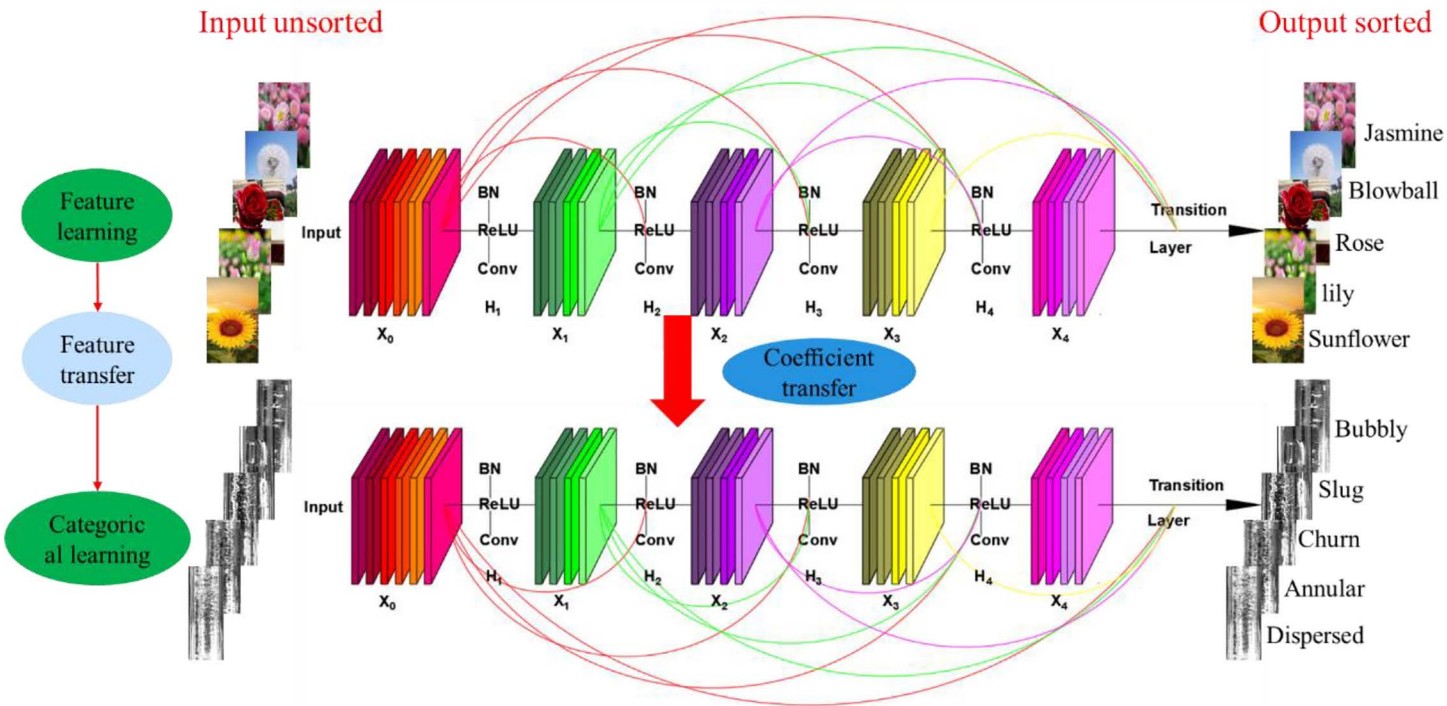

**Fig 11. Schematic diagram of flow pattern recognition via transfer learning.**

## 3.1 Ablation study and transfer learning analysis of ED-DenseNet

To evaluate the contributions of individual components in the ED-DenseNet model and analyze the impact of transfer learning on model performance, this study conducted five ablation experiments under identical experimental conditions. These experiments are as follows: (1) the original DenseNet121 model trained from scratch; (2) DenseNet121 integrated with dilated convolution (DC); (3) DenseNet121 integrated with the ECA attention mechanism; (4) the complete ED-DenseNet architecture (combining ECA attention mechanism and dilated convolution) trained using transfer learning; and (5) the complete ED-DenseNet architecture trained entirely from scratch without transfer learning. Table 2 shows the experimental results for these configurations on the gas-liquid two-phase flow dataset.

The experimental results presented in Table 2 verify that each component proposed in this paper effectively enhances the classification performance of the model. Comparing DenseNet121 and DC-DenseNet121, incorporating dilated convolution improved the classification accuracy by 1.13%. This indicates that dilated convolutions effectively expand the receptive field and enhance multi-scale feature extraction, allowing better capture of complex flow pattern features without significantly

**Table 2. Ablation study results of the ED-DenseNet model.**

| Model | Accuracy | Precision | Recall | F1 Score |
|---|---|---|---|---|
| DenseNet121 | 95.26% | 94.37% | 93.12% | 93.74% |
| DC-DenseNet121 | 96.39% | 95.62% | 95.27% | 95.44% |
| ECA-DenseNet121 | 97.63% | 96.31% | 95.82% | 96.06% |
| ED-DenseNet (with Transfer Learning) | 97.82% | 96.86% | 97.63% | 97.24% |
| ED-DenseNet (without Transfer Learning) | 94.89% | 93.52% | 92.74% | 93.12% |

increasing computational complexity. The comparison between DenseNet121 and ECA-DenseNet121 indicates that integrating the ECA attention mechanism alone improved the accuracy by 2.37% and the recall by 2.7%, demonstrating that the attention mechanism significantly improves the model's ability to highlight crucial feature channels, thus enhancing classification stability and robustness. Furthermore, when the ECA attention mechanism and dilated convolution were combined in the ED-DenseNet model, the accuracy reached the highest value of 97.82%, further validating the synergistic effect of these two mechanisms in effectively enhancing the recognition performance of gas-liquid two-phase flow patterns.

Additionally, the ED-DenseNet model without transfer learning showed significantly lower performance (accuracy of 94.89%) compared to the version utilizing transfer learning (97.82%), a difference of 2.93%, along with a substantial increase in training time (from 255 min to 348 min). Specifically, without transfer learning, the ED-DenseNet model even underperformed compared to models employing only dilated convolutions (DC-DenseNet121: 96.39%) or the ECA mechanism (ECA-DenseNet121, 97.63%). This phenomenon may be due to ED-DenseNet having more parameters compared to single-component models. Without proper initialization provided by transfer learning, the ED-DenseNet is susceptible to gradient instability or vanishing gradients in the early training phases, negatively impacting convergence speed and final performance. Additionally, deep models trained from scratch on small datasets are more prone to overfitting, while transfer learning can effectively mitigate overfitting and enhance generalization.

In summary, the introduction of the ECA attention mechanism and dilated convolution significantly enhances the feature extraction capability of ED-DenseNet. Their combined effects not only optimize classification accuracy but also improve generalization performance. Furthermore, transfer learning plays a crucial role in reducing overfitting, improving model generalization, and significantly decreasing training time. These findings collectively confirm the effectiveness of the proposed ED-DenseNet architecture and transfer learning strategy for gas-liquid two-phase flow pattern classification.

### 3.2 ED-DenseNet classification performance

To validate the classification capability of ED-DenseNet compared to DenseNet121, both models were trained under the same experimental conditions, and their loss rates and classification accuracy on the gas-liquid two-phase flow dataset were compared. The specific results are shown in Figs 12 and 13. From Fig 12, it can be observed that as the number of training iterations increases, the loss rates of both ED-DenseNet and DenseNet121 decrease, but ED-DenseNet exhibits

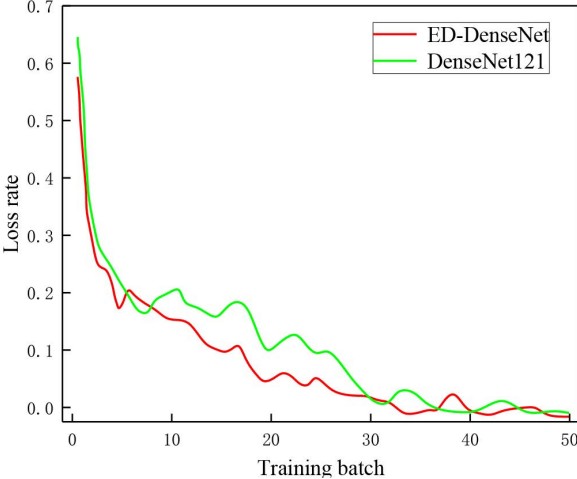

**Fig 12. Loss rate change curve.** (The loss and accuracy curves in Figs 11 and 12 remain unchanged, as the additional transitional flow dataset is only used in the test phase and is not included in training or validation).

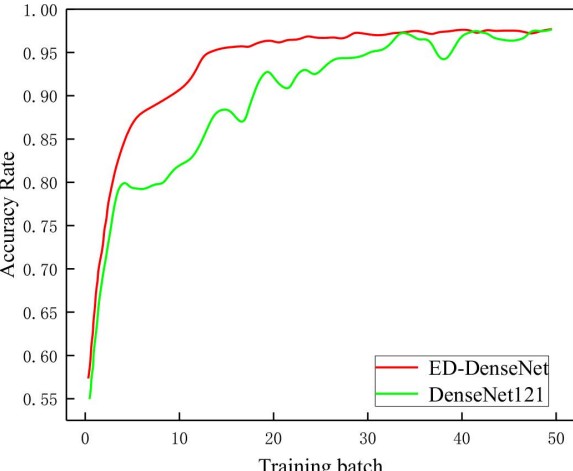

**Fig 13. Accuracy change curve.**

a faster decline, indicating its superior learning ability. Fig 13 shows that when the training epochs reach 20, the accuracy of ED-DenseNet quickly stabilizes, while DenseNet121 continues to improve gradually. These results indicate that the proposed method in this study demonstrates advantages in terms of training stability, model convergence speed, and generalization capability, ultimately achieving superior classification performance.

To evaluate the model's classification capability for the five primary gas-liquid two-phase flow patterns, 460 samples per flow pattern, totaling 2,300 images, were selected from the test set. Fig 14 presents the confusion matrix of ED-DenseNet on the primary flow pattern test set. The results show that the model achieved 100% accuracy for Bubbly Flow and Slug Flow, indicating that these flow patterns have clear gas-liquid interfaces and are easily distinguishable. However, Churn Flow, Annular Flow, and Dispersed Flow exhibited some misclassifications due to unstable gas-liquid interfaces and high turbulence intensity.

To further assess the model's generalization capability in transitional flow patterns (Slug→Churn, Churn→Annular), we introduced an additional Transitional Flow Test Set, consisting of 1,000 transitional flow images, to evaluate the model's performance under ambiguous flow boundaries. These samples simulate real-world engineering applications where gradual flow transitions occur, but due to their blurred boundaries, classification remains challenging. The classification performance results are presented in Table 3. In the transitional flow dataset, the model's accuracy slightly declined: Slug→Churn (89.42%), primarily due to the increased turbulence intensity as Taylor bubbles break up, making flow characteristics highly dynamic. Churn→Annular (91.07%), where the liquid film has not yet fully stabilized, resulted in some samples being misclassified as either Churn Flow or Annular Flow.

Although ED-DenseNet achieved excellent overall classification accuracy, certain complex flow patterns (Annular Flow, Churn Flow, Dispersed Flow) and transitional flow patterns (Slug→Churn, Churn→Annular) still exhibited some misclassification cases. Below, we provide an in-depth analysis of these cases based on misclassification statistics:

- Annular Flow: 12 samples misclassified as Churn Flow (2.6%). The thin liquid film in annular flow makes gas-phase feature extraction difficult. During the Churn→Annular transition, the liquid film is not yet fully stable, and some samples still exhibit churn-like characteristics, leading to misclassification.

- Churn Flow: 14 samples misclassified as Annular Flow (3.0%), 6 samples misclassified as Dispersed Flow (1.3%). Churn flow has highly unstable turbulence, causing significant variations in flow features across samples. Some samples exhibit liquid film characteristics similar to annular flow, while others share characteristics with dispersed flow, leading to classification errors.

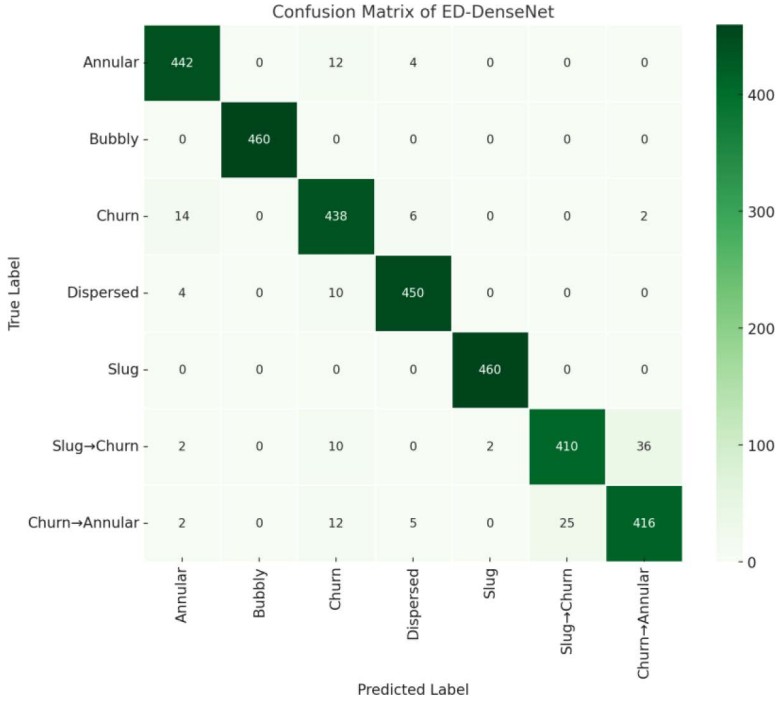

**Fig 14. Confusion matrix for flow pattern classification.**

**Table 3. Classification performance of primary and transitional flow patterns.**

| Flow pattern classification | Accuracy | Recall | Precision | F1 Score |
|---|---|---|---|---|
| Annular | 96.09% | 96.52% | 96.01% | 96.26% |
| Bubbly | 100% | 100% | 100% | 100% |
| Churn | 95.21% | 95.36% | 96.21% | 95.78% |
| Dispersed | 97.82% | 97.96% | 97.62% | 97.79% |
| Slug | 100% | 100% | 100% | 100% |
| Slug→Churn | 89.42% | 87.63% | 88.91% | 88.26% |
| Churn→ Annular | 91.07% | 89.48% | 90.32% | 89.90% |

- Slug→Churn Transitional Flow: 10 samples misclassified as Churn Flow (2.2%). As Taylor bubbles break up during the Slug→Churn transition, turbulence intensity increases, making flow structures more unstable. This leads to some samples being misclassified as Churn Flow.

- Churn→Annular Transitional Flow: 25 samples misclassified as Annular Flow (5.4%). The Churn→Annular transition involves the gradual formation of a liquid film, but some samples still retain churn-like features. During feature extraction, the model may misclassify such samples as Annular Flow.

The experimental results validate the effectiveness and robustness of ED-DenseNet in gas-liquid two-phase flow pattern recognition. The model achieves an accuracy above 95% for all primary flow patterns, demonstrating superior classification performance. Additionally, transitional flow patterns achieve accuracy above 89%, proving that the model generalizes well during flow transitions. The primary cause of misclassification is the gradual nature of flow transitions, further confirming that flow pattern gradual transformation is a key factor affecting classification accuracy.

### 3.3 Comparative experiments with similar models

To further validate the superiority of the proposed ED-DenseNet model for gas-liquid two-phase flow pattern classification, comparative experiments were conducted under the same training environment and optimization settings against MobileNet V3 [33], ConvNeXt [34], ShuffleNet V2 [35], and the Flow-Hilbert–CNN model [17].

All models were implemented based on the PyTorch framework and trained for 50 epochs using the same gas-liquid two-phase flow image dataset. The classification accuracies of five primary flow patterns (Bubbly Flow, Slug Flow, Churn Flow, Annular Flow, Dispersed Flow) were evaluated, as illustrated in Fig 15. It can be observed from Fig 15 that Bubbly Flow, characterized by discrete small bubbles distributed in the liquid phase with clear interfaces, exhibited the highest recognition accuracy (up to 100%) across all models. Slug Flow, developed from Bubbly Flow and also featuring clear interfaces, similarly demonstrated high accuracy. In contrast, due to the increased gas phase fraction, the Annular Flow, Churn Flow, and Dispersed Flow patterns presented more continuous phase distributions and indistinct interfaces, increasing the difficulty in feature extraction and leading to lower recognition accuracies. Among these, ConvNeXt and ShuffleNet V2 performed the worst in identifying the complex flow patterns (Annular Flow, Churn Flow, and Dispersed Flow), with their lowest accuracies

The overall training performances of these models are summarized in Table 4. The ShuffleNet V2 model had the shortest training time (161 min), but also the lowest overall accuracy (82.21%), possibly due to its Channel Grouping and Shuffle operations, which reduce computational complexity but restrict information flow between channels, limiting the model's ability to capture detailed flow features. ConvNeXt, utilizing Large-Kernel Convolutions, Layer Normalization, and

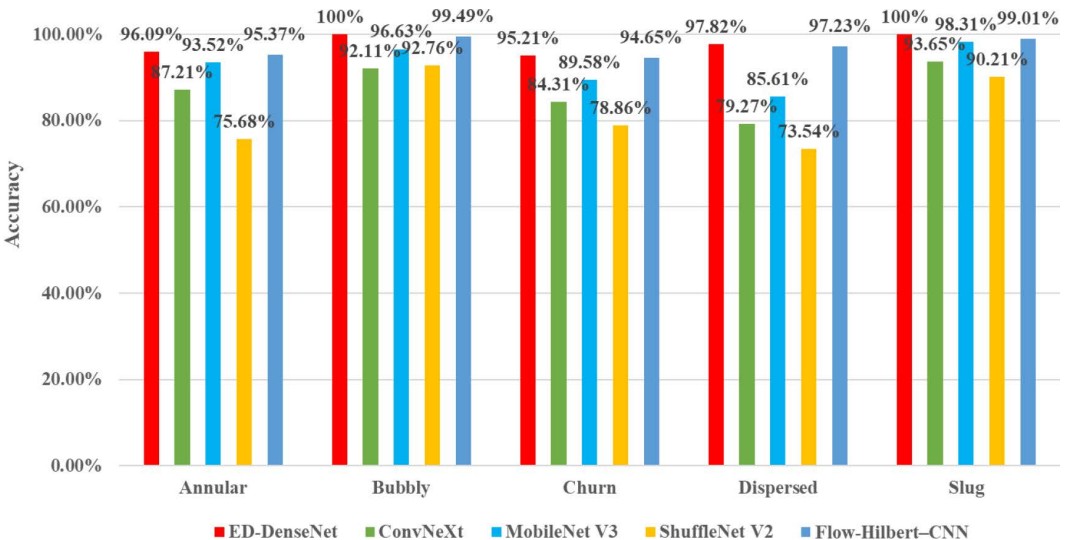

**Fig 15. Deep learning model flow pattern recognition accuracy.**

**Table 4. Performance and training efficiency of different models.**

| Model | Batch-size | Accuracy | Recall | Precision | F1 Score | Training Time |
|---|---|---|---|---|---|---|
| ED-DenseNet | 32 | 97.82% | 96.86% | 97.63% | 97.24% | 255 min |
| MobileNet V3 | 32 | 92.73% | 91.43% | 93.24% | 92.33% | 188 min |
| ConvNeXt | 32 | 87.31% | 85.26% | 87.96% | 86.59% | 389 min |
| ShuffleNet V2 | 32 | 82.21% | 81.38% | 83.12% | 82.24% | 161 min |
| Flow-Hilbert–CNN | 32 | 97.15% | 95.42% | 96.83% | 96.12% | 310 min |

multi-scale feature fusion, had the longest training time (389 min) but achieved only 87.31% accuracy. This result suggests that its architecture was less effective in capturing the features of gas-liquid two-phase flow. MobileNet V3, benefiting from Depthwise Separable Convolutions and AutoML-based architecture optimization, achieved relatively high accuracy (92.73%) with a short training duration (188 min), making it particularly suitable for resource-constrained applications.

To provide a comprehensive evaluation of the proposed ED-DenseNet model, the Flow-Hilbert–CNN method proposed by Zhang et al. [17] was reproduced in this study as one of the baseline models for comparison. It is important to note that the original Flow-Hilbert–CNN model was designed based on one-dimensional water mass signal (WMS) data, which are encoded into two-dimensional feature maps using Hilbert curve-based encoding before being fed into a CNN classifier. However, our research focuses on static image-based gas-liquid two-phase flow pattern recognition. Therefore, in this study, the core CNN architecture and parameter settings of the Flow-Hilbert–CNN model were retained, but instead of using WMS signals, we applied the model directly to our self-collected static image dataset. No WMS-related preprocessing or signal data were involved.

This approach allows a fair performance comparison while ensuring that the application context and data sources are distinct, thus avoiding any infringement or inappropriate use of proprietary data. The adaptation demonstrates how the Flow-Hilbert–CNN model performs when applied to static flow pattern images, providing valuable insight into its effectiveness outside of its original signal-based domain. The overall accuracy achieved by Flow-Hilbert–CNN on the gas-liquid two-phase flow dataset was 97.15%, close to that of ED-DenseNet (97.82%). Specifically, Flow-Hilbert–CNN achieved high accuracies of 99.49% and 99.01% in stable flow patterns with distinct features, such as Bubbly Flow and Slug Flow. However, its accuracy was notably lower for Annular Flow (95.37%), Churn Flow (94.65%), and Dispersed Flow (97.23%), which are characterized by stronger turbulence and irregular interfaces. The primary reason is that the Flow-Hilbert–CNN relies heavily on temporal information, whereas this study's dataset consists mainly of static images, restricting its effectiveness in single-image classification tasks. The Hilbert transform primarily converts temporal features to frequency-domain features, which may be limited in capturing the characteristics of highly turbulent and irregular-interface flow patterns. In contrast, ED-DenseNet, by employing the ECA attention mechanism and dilated convolutions, can more effectively extract multi-scale spatial features, thus achieving superior generalization and accuracy.

Additionally, to clearly illustrate the performance differences among models during training, we recorded the accuracy and loss curves for each model across different epochs, as shown in Figs 16 and 17. It can be observed that due to their lightweight architectures, ShuffleNet V2 and MobileNet V3 achieved the fastest improvement in accuracy and the most rapid decline in loss during the early stages of training. However, their performance plateaued after approximately 8 epochs. The ConvNeXt model exhibited a slower improvement in accuracy and reduction in loss, which may be attributed to the inefficiency of large convolutional kernels in extracting gas-liquid two-phase flow features. Nevertheless, ConvNeXt reached near-peak accuracy after around 35 epochs, surpassing the previously better-performing ShuffleNet V2. Although ED-DenseNet demonstrated relatively moderate performance at the beginning, it exhibited superior feature learning capability, maintaining a steady upward trend even when other models stabilized. Notably, ED-DenseNet achieved over 97% accuracy within 20 epochs, whereas the original DenseNet121 reached stable performance only after approximately 45 epochs. These results further confirm the synergistic effect of integrating the ECA attention mechanism and dilated convolutions in ED-DenseNet, effectively enhancing feature extraction and generalization performance in gas-liquid two-phase flow pattern classification.

In summary, the ED-DenseNet demonstrated clear advantages in terms of accuracy, feature extraction capabilities, and computational efficiency in gas-liquid two-phase flow pattern recognition tasks. It particularly excelled in complex flow pattern identification compared to other models, making it suitable for practical industrial applications. MobileNet V3 represents an optimal choice for resource-limited scenarios. ConvNeXt and ShuffleNet V2 exhibited comparatively lower performance due to architectural or computational limitations. Moreover, the comparative results of Flow-Hilbert–CNN further

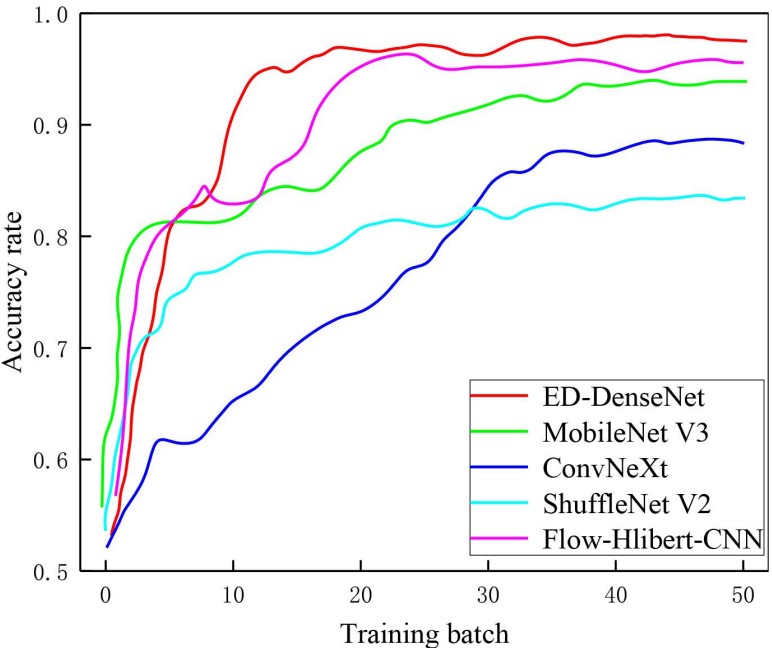

**Fig 16. Model accuracy curves.**

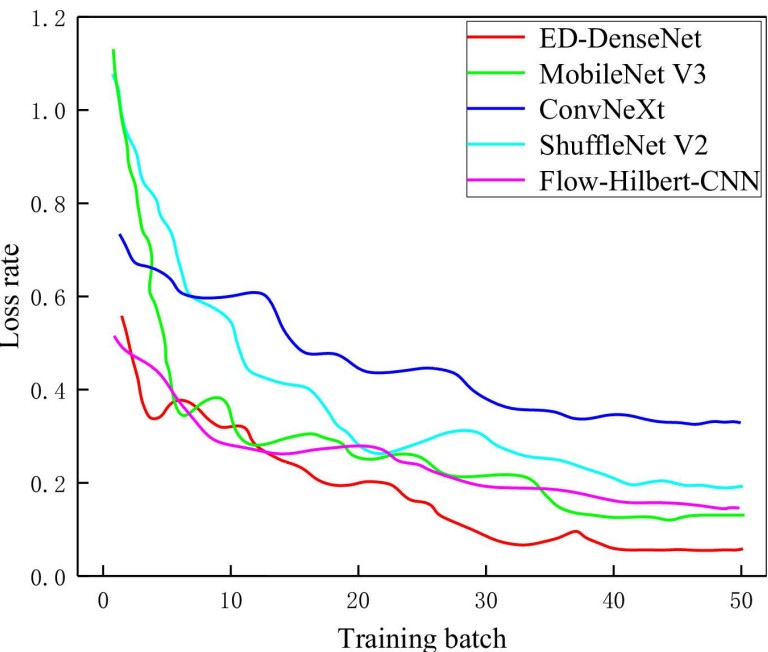

**Fig 17. Model loss curves.**

reinforce the effectiveness of attention mechanisms and dilated convolution methods incorporated into ED-DenseNet for the task of gas-liquid two-phase flow pattern recognition.

## 3.4 Optimizer comparison experiment

To explore the adaptability of different optimizers to various model architectures, this study evaluates the performance of SGD, Adam, AdaMax, and Radam [36] on ED-DenseNet, DenseNet121, MobileNetV3, and ConvNeXt. The complexity and depth characteristics of different models determine the suitability of the optimizers. The training, validation, and test set classification accuracy for each optimizer-model combination are summarized in Tables 5 and 6.

Table 5 presents the classification performance of different optimizers on ED-DenseNet. The results indicate that Adam achieves the highest test accuracy of 96.71%, making it the most suitable optimizer for this model. In contrast, SGD and RAdam exhibit inferior performance, with test accuracies of 87.26% and 90.29%, respectively, demonstrating their weaker adaptability to ED-DenseNet. AdaMax achieves a slightly lower training accuracy than Adam, but it exhibits a higher validation accuracy, suggesting that AdaMax may provide better generalization in certain scenarios.

Table 6 further illustrates the performance of different optimizers across DenseNet121, MobileNetV3, and ConvNeXt. Adam consistently achieves the highest test accuracy in DenseNet121 and MobileNetV3, confirming its superior optimization capabilities for relatively shallow models. However, in ConvNeXt, AdaMax surpasses Adam in validation accuracy, suggesting that AdaMax may offer better generalization in complex networks. Nevertheless, Adam maintains high stability on ConvNeXt, achieving a competitive test accuracy of 95.24%, indicating its robustness across various deep learning tasks.

**Table 5. Comparison of different optimizers on ED-DenseNet.**

| Model | Training set accuracy | Validation set accuracy | Test set accuracy |
|---|---|---|---|
| ED-DenseNet (SGD) | 94.53% | 92.31% | 87.26% |
| ED-DenseNet (Radam) | 96.31% | 92.43% | 90.29% |
| ED-DenseNet (Adam) | 97.82% | 95.33% | 96.71% |
| ED-DenseNet (AdaMax) | 97.12% | 97.01% | 96.31% |

**Table 6. Comparison of different optimizers on DenseNet121, MobileNetV3, and ConvNeXt.**

| Model | Training set accuracy | Validation set accuracy | Test set accuracy |
|---|---|---|---|
| DenseNet121 (SGD) | 95.26% | 93.74% | 95.12% |
| DenseNet121 (RAdam) | 96.47% | 94.32% | 95.68% |
| DenseNet121 (Adam) | 97.34% | 95.81% | 96.74% |
| DenseNet121 (AdaMax) | 97.18% | 95.92% | 96.58% |
| MobileNetV3 (SGD) | 95.62% | 91.58% | 89.02% |
| MobileNetV3 (RAdam) | 96.47% | 93.12% | 91.74% |
| MobileNetV3 (Adam) | 97.34% | 95.22% | 95.61% |
| MobileNetV3 (AdaMax) | 97.18% | 96.04% | 96.02% |
| ConvNeXt (SGD) | 94.31% | 90.74% | 88.14% |
| ConvNeXt (RAdam) | 95.86% | 92.53% | 91.02% |
| ConvNeXt (Adam) | 96.93% | 95.47% | 95.24% |
| ConvNeXt (AdaMax) | 96.81% | 96.32% | 95.96% |

The adaptability of an optimizer depends on the depth and structural complexity of the network. In deeper networks such as ED-DenseNet and ConvNeXt, the fixed learning rate of SGD struggles to optimize complex parameters efficiently, often leading to slow convergence, vanishing gradients, or oscillations, which significantly reduce its test accuracy compared to Adam and AdaMax. In contrast, Adam's adaptive learning rate mechanism enhances stability in shallower models like DenseNet121 and MobileNetV3, making it the most suitable optimizer for these architectures. Compared to Adam, AdaMax demonstrates superior weight adjustment stability in deeper networks like ConvNeXt, reducing gradient fluctuations and thereby improving model generalization. While AdaMax achieves a higher validation accuracy than Adam, Adam still maintains a strong test accuracy, confirming its continued relevance for optimizing deep networks.

SGD performs better on DenseNet121 (95.12% test accuracy) compared to deeper models, suggesting that it remains a viable option for relatively shallow architectures. However, its performance drops significantly in ED-DenseNet and ConvNeXt, reinforcing its limitations in deep learning tasks. RAdam consistently underperforms compared to Adam and AdaMax across all models, indicating that its warm-up mechanism does not provide significant benefits in this particular task.

Experimental results indicate that AdaMax outperforms Adam only in deeper and more complex models such as ConvNeXt, but not across all architectures. This aligns with AdaMax's theoretical advantage in stabilizing training under large gradient variations. Adam remains the most reliable optimizer for most scenarios, particularly in relatively shallow architectures such as DenseNet121 and MobileNetV3, where it consistently delivers superior performance. While SGD demonstrates moderate effectiveness in DenseNet121, its optimization ability deteriorates in deeper networks. Therefore, optimizer selection should be tailored to the model architecture and task requirements. Adam and AdaMax are more suitable for optimizing deep learning models, whereas SGD remains viable for shallow networks or tasks requiring strong regularization.

### 3.5 Model generalizability comparison

To validate the generalizability of the method proposed in this paper, a dataset of two-phase flow pattern images from a horizontal pipe (consisting of 1050 images) involving nitrogen condensation was selected for the study. This was done to avoid irrelevant features unrelated to the flow conditions, which differ from those in the vertical pipe examined in this paper. Four typical flow patterns (Bubbly Flow et al.) were selected from these images through manual segmentation. Each image is assumed to represent one type of flow pattern, with the dataset split into a training set, validation set, and test set in a 7:2:1 ratio. Fig 18 displays representative images for each flow pattern.

To further validate the overall performance of the proposed model, a comparative experiment was conducted on this dataset between the method presented in this paper and the MobileNet V3、ConvNeXt and ShuffleNet V2 models. The experimental results are shown in Table 7. As can be seen from the table, the training accuracy of the proposed method on the new dataset is 92.62%, which is higher than that of other similarly excellent convolutional neural networks. This indicates that the proposed method possesses strong generalizability and robustness in generalization.

### 4. Conclusion

Accurately identifying two-phase flow patterns is crucial for studying heat transfer and flow characteristics. With technological advancements, quantitative flow pattern recognition based on two-phase flow images has become a significant research trend. This paper leverages the advantages of Convolutional Neural Networks (CNNs) and attention mechanisms in image classification and feature extraction, proposing a flow pattern recognition method based on ED-DenseNet. A dataset containing five typical flow patterns—Annular, Bubbly, Churn, Dispersed, and Slug—was constructed, along with an additional transitional flow pattern test set to evaluate the robustness of the model. The main conclusions of this paper are summarized as follows:

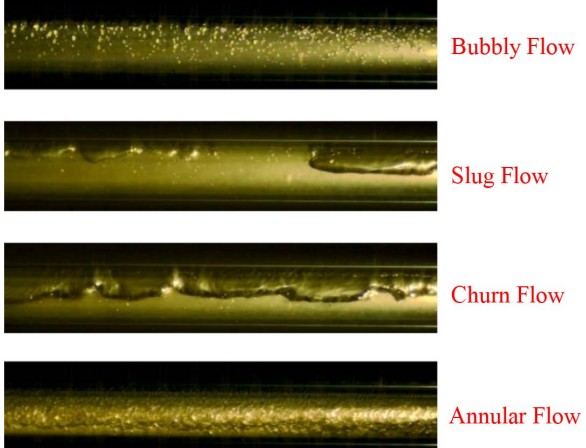

Bubbly Flow

Slug Flow

Churn Flow

Annular Flow

**Fig 18. Two-phase flow patterns for nitrogen condensation.**

**Table 7. Generalization experiment results.**

| Model | Training set accuracy | Validation set accuracy | Test set accuracy |
|---|---|---|---|
| ED-DenseNet | 92.62% | 91.11% | 88.26% |
| MobileNet V3 | 88.46% | 86.93% | 85.31% |
| ConvNeXt | 86.37% | 85.62% | 85.12% |
| ShuffleNet V2 | 84.26% | 83.33% | 81.26% |

(1) The integration of the ECA attention mechanism and dilated convolution module into the DenseNet121 network enhances channel information awareness, enabling multi-layer feature extraction and enhancement, thereby improving the accuracy of flow pattern recognition.

(2) Compared with MobileNet V3, ConvNeXt, ShuffleNet V2, and Flow-Hilbert–CNN models, the proposed ED-DenseNet exhibits superior performance, particularly in recognizing complex and transitional flow patterns, with an overall recognition rate of 97.82% and a moderate training time (255 minutes), making it the most suitable method for flow pattern recognition tasks.

(3) The adoption of transfer learning effectively improves the model's convergence speed and generalization ability. The accuracy and loss values stabilize within 50 epochs, alleviating the overfitting problem on small sample datasets.

(4) The proposed method was further validated on a two-phase flow pattern image dataset involving nitrogen condensation in a horizontal pipe, demonstrating superior generalization and versatility compared to other models.

## Acknowledgments

We thank Feng Nie for providing the two-phase flow pattern image dataset for nitrogen condensation.

## Author contributions

**Conceptualization:** Jie Liu, Yang Wu.

**Data curation:** Yang Wu.

**Formal analysis:** Yang Wu.

**Funding acquisition:** Yang Wu.

**Investigation:** Jie Liu.

**Methodology:** Jie Liu.

**Project administration:** Yang Wu.

**Resources:** Jie Liu.

**Software:** Yang Wu.

**Supervision:** Yang Wu.

**Validation:** Jie Liu.

**Visualization:** Yang Wu.

**Writing – original draft:** Jie Liu.

**Writing – review & editing:** Jie Liu, Yang Wu.

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
