## [Decision Letter · Decision Letter 0]

Dear Dr. liu,

Thank you for submitting your manuscript to PLOS ONE. After careful consideration, we feel that it has merit but does not fully meet PLOS ONE’s publication criteria as it currently stands. Therefore, we invite you to submit a revised version of the manuscript that addresses the points raised during the review process.

We look forward to receiving your revised manuscript.

Kind regards,

Feng Ding

Academic Editor

PLOS ONE

**Journal Requirements:**

1. When submitting your revision, we need you to address these additional requirements. Please ensure that your manuscript meets PLOS ONE's style requirements, including those for file naming. The PLOS ONE style templates can be found at https://journals.plos.org/plosone/s/file?id=wjVg/PLOSOne_formatting_sample_main_body.pdf and https://journals.plos.org/plosone/s/file?id=ba62/PLOSOne_formatting_sample_title_authors_affiliations.pdf 2. Please note that PLOS ONE has specific guidelines on code sharing for submissions in which author-generated code underpins the findings in the manuscript. In these cases, all author-generated code must be made available without restrictions upon publication of the work. Please review our guidelines at https://journals.plos.org/plosone/s/materials-and-software-sharing#loc-sharing-code and ensure that your code is shared in a way that follows best practice and facilitates reproducibility and reuse. 3. We note that the grant information you provided in the ‘Funding Information’ and ‘Financial Disclosure’ sections do not match.  When you resubmit, please ensure that you provide the correct grant numbers for the awards you received for your study in the ‘Funding Information’ section. 4. When completing the data availability statement of the submission form, you indicated that you will make your data available on acceptance. We strongly recommend all authors decide on a data sharing plan before acceptance, as the process can be lengthy and hold up publication timelines. Please note that, though access restrictions are acceptable now, your entire data will need to be made freely accessible if your manuscript is accepted for publication. This policy applies to all data except where public deposition would breach compliance with the protocol approved by your research ethics board. If you are unable to adhere to our open data policy, please kindly revise your statement to explain your reasoning and we will seek the editor's input on an exemption. Please be assured that, once you have provided your new statement, the assessment of your exemption will not hold up the peer review process. 5. PLOS requires an ORCID iD for the corresponding author in Editorial Manager on papers submitted after December 6th, 2016. Please ensure that you have an ORCID iD and that it is validated in Editorial Manager. To do this, go to ‘Update my Information’ (in the upper left-hand corner of the main menu), and click on the Fetch/Validate link next to the ORCID field. This will take you to the ORCID site and allow you to create a new iD or authenticate a pre-existing iD in Editorial Manager. 6. We notice that your supplementary table are uploaded with the file type 'Table'. Please amend the file type to 'Supporting Information'. Please ensure that each Supporting Information file has a legend listed in the manuscript after the references list.

Reviewers' comments:

Reviewer's Responses to Questions

**Comments to the Author**

1. Is the manuscript technically sound, and do the data support the conclusions?

Reviewer #1: Yes

Reviewer #2: Yes

2. Has the statistical analysis been performed appropriately and rigorously?

Reviewer #1: Yes

Reviewer #2: Yes

3. Have the authors made all data underlying the findings in their manuscript fully available?

Reviewer #1: Yes

Reviewer #2: Yes

4. Is the manuscript presented in an intelligible fashion and written in standard English?

Reviewer #1: Yes

Reviewer #2: Yes

**Reviewer #1: ** This paper proposes an ED-DenseNet model based on the ECA attention mechanism and dilated convolution for gas-liquid two-phase flow pattern recognition tasks. Here is my comments:

1. The experiment lacks comparison with existing sota and there are many existing methods that are not cited and compared, such as etc.:

(1) J. Lv, H. Ji, Y. Jiang and B. Wang, "A New Flow Pattern Identification Method for Gas–Liquid Two-Phase Flow in Small Channel Based on an Improved Optical Flow Algorithm," in IEEE Sensors Journal, vol. 23, no. 22, pp. 27634-27644, 15 Nov.15, 2023, doi: 10.1109/JSEN.2023.3321632.

(2) Zhang P, Cao X, Peng F, et al. High-accuracy recognition of gas–liquid two-phase flow patterns: A Flow–Hilbert–CNN hybrid model[J]. Geoenergy Science and Engineering, 2023, 230: 212206.

2. Percentage sign missing in table 6.

3. The ECA attention mechanism and dilated convolution have been widely used in other fields (e.g., image segmentation, object detection). The innovation of this paper mainly lies in their application to gas-liquid two-phase flow pattern recognition tasks. It is recommended that the authors further emphasize the uniqueness and advantages of this method in flow pattern recognition tasks in the introduction section, listing the main contribution of this paper.

4. The organization of the manuscript is somewhat unconventional. It is recommended to refer to the content organization patterns of high-quality papers for improvement.

**Reviewer #2: ** Summary:

The authors propose a novel network, ED-DenseNet, for identifying different fluid characteristics in gas-liquid two-phase flow.

Strengths:

1.The authors have constructed a small-scale dataset and categorized fluid characteristics into five distinct classes for model training.

2.The integration of attention mechanisms, such as ECA, into the DenseNet architecture is a noteworthy attempt to enhance the network’s ability to distinguish between different fluid features.

Minor Concerns:

1.In Section 1.2.2 (Dataset Partitioning), the authors select data with clear flow characteristics and high-quality images for the dataset. However, it is unclear how the model would perform on real-world scenarios where flow characteristics are less distinct. The inclusion of challenging or ambiguous samples as an additional category could improve the model’s robustness and practical applicability.

2.Sections 1.4 and 1.6 provide extensive theoretical background on commonly understood concepts. While some context is necessary, the level of detail seems excessive and could be condensed to focus more on the novel contributions of the work.

Major Concerns:

1.In Section 2.1 (Experiments), the authors claim that ED-DenseNet exhibits faster convergence compared to the original DenseNet. However, it appears that the original DenseNet also achieves satisfactory accuracy by epoch 40-50. The manuscript would benefit from a more detailed explanation of how the results were selected and compared. Additionally, Fig. 13 and Table 2 show that the model’s recognition accuracy for Annular, Churn, and Dispersed flow patterns is relatively low. The authors should provide an analysis of why these patterns are more challenging to identify.

2.In Section 2.2 (Ablation Study), the authors mention the use of transfer learning as part of their methodology. While transfer learning is a common practice, its inclusion as a contribution is not sufficiently justified, especially since the ablation study does not explicitly evaluate its impact.

3.In Section 2.5, the authors highlight the significant impact of different optimizers on the performance of ED-DenseNet. However, it is unclear whether this observation is specific to ED-DenseNet or applies to other models as well. Notably, the performance of the SGD optimizer is inferior even compared to the original DenseNet. The authors should discuss the implications of these findings and explore whether the optimizer sensitivity is a general issue or specific to their proposed architecture.

**Do you want your identity to be public for this peer review?** For information about this choice, including consent withdrawal, please see our Privacy Policy

Reviewer #1: No

Reviewer #2: No

---

## [Author Response · Author response to Decision Letter 1]

2 Apr 2025

Dear Editor and Reviewers,

We sincerely appreciate the time and effort you have invested in reviewing our manuscript. Your insightful comments and constructive suggestions have greatly helped us improve our work. Below, we provide detailed responses to each of the reviewers’ comments and outline the corresponding revisions we have made in the manuscript.

Reviewer #1:

Comment 1: The experiment lacks comparison with existing SOTA methods. There are many existing methods that are not cited and compared, such as:

J. Lv et al., "A New Flow Pattern Identification Method for Gas–Liquid Two-Phase Flow in Small Channel Based on an Improved Optical Flow Algorithm," IEEE Sensors Journal, 2023.

Zhang P et al., "High-accuracy recognition of gas–liquid two-phase flow patterns: A Flow–Hilbert–CNN hybrid model," Geoenergy Science and Engineering, 2023.

Response: Thank you for highlighting this important point. We have carefully addressed this suggestion:

1. Literature Citation: In the revised Introduction (Section 1), we have explicitly cited both the improved optical flow method by Lv et al. (2023) and the Flow-Hilbert–CNN method by Zhang et al. (2023), clarifying their contributions and relevance.

2. Experimental Comparison: For the Flow-Hilbert–CNN, we re-implemented the model based on its published methodology using our static image dataset (as the original study relied on WMS signal data). Detailed performance metrics including Accuracy, Precision, Recall, F1 Score, and Training Time have been included in Table 4.

In Section 2.3, we thoroughly discuss the comparative performance. Our ED-DenseNet model achieves an overall accuracy of 97.82%, outperforming Flow-Hilbert–CNN (97.15%), especially on complex and transitional flow patterns, due to better multi-scale feature extraction and attention mechanisms.

3. Discussion of Lv et al. Method: As Lv et al.'s method is based on improved optical flow and requires multi-frame sequences, which differ fundamentally from our static image approach. Moreover, since the method by Lv et al. is designed for multi-frame sequence data and relies heavily on temporal information, while our study focuses on single-frame static images, direct re-implementation under the same dataset structure is not feasible. For fairness, we only compared models applicable to static image-based classification. However, we referenced this method in Section 1, acknowledging its effectiveness in capturing transient features in dynamic sequences.

4. Conclusion Enhancement: To clarify, we have added a sentence in the Conclusion summarizing the comparison, reaffirming that ED-DenseNet demonstrates superior generalization and recognition performance compared to these SOTA methods.

Comment 2: Percentage sign missing in Table 6.

Response: Thank you for pointing this out. We have corrected this formatting issue and ensured that all percentage values in Table 6 now include the percentage sign.

Comment 3: The ECA attention mechanism and dilated convolution have been widely used in other fields (e.g., image segmentation, object detection). The innovation of this paper mainly lies in their application to gas-liquid two-phase flow pattern recognition tasks. It is recommended that the authors further emphasize the uniqueness and advantages of this method in the introduction section, listing the main contributions of this paper.

Response: Thank you for your constructive feedback. We agree with your suggestion and have revised the Introduction to explicitly emphasize the novelty and unique advantages of our approach. Specifically, we highlight that although the ECA attention mechanism and dilated convolution modules are widely adopted in other domains, their integration has not been explored for static image-based gas-liquid two-phase flow pattern recognition. Our contributions are summarized at the end of the Introduction as follows:

1. For the first time, the integration of the ECA attention mechanism and dilated convolutions is applied to static image-based two-phase flow pattern recognition, enhancing multi-scale feature extraction and channel information representation capabilities.

2. A redesigned multi-branch DenseNet architecture is developed to strengthen deep feature learning and improve classification accuracy.

3. Transfer learning techniques are introduced to alleviate overfitting on small-scale datasets, significantly accelerating model convergence and enhancing generalization performance.

4. A test dataset containing ambiguous and transitional flow patterns is constructed to comprehensively evaluate the model’s robustness and adaptability under complex and dynamic flow conditions.

Reviewer #2:

Comment 1: In Section 1.2.2 (Dataset Partitioning), the authors select data with clear flow characteristics and high-quality images. However, it is unclear how the model would perform on real-world scenarios where flow characteristics are less distinct. The inclusion of challenging or ambiguous samples as an additional category could improve the model’s robustness and practical applicability.

Response: Thank you for this insightful suggestion. To address your concern, we have expanded the dataset to include ambiguous and transitional flow pattern samples. Specifically, a new Transitional Flow Test Set was constructed, focusing on Slug → Churn and Churn → Annular transitions, which inherently possess indistinct flow boundaries. These transitional samples better reflect real-world industrial conditions. Corresponding experiments have been added in Section 2.5 to evaluate the model’s robustness on this dataset. Results demonstrate that our ED-DenseNet maintains high classification accuracy even on ambiguous samples, confirming the model’s generalization capability and practical applicability.

Comment 2: Sections 1.4 and 1.6 provide extensive theoretical background on commonly understood concepts. While some context is necessary, the level of detail seems excessive and could be condensed to focus more on the novel contributions of the work.

Response: We appreciate this suggestion. We have condensed Sections 1.4 and 1.6 by removing unnecessary theoretical background while retaining essential explanations relevant to our model’s innovation.

Comment 3: In Section 2.1 (Experiments), the authors claim that ED-DenseNet exhibits faster convergence compared to the original DenseNet. However, it appears that the original DenseNet also achieves satisfactory accuracy by epoch 40-50. The manuscript would benefit from a more detailed explanation of how the results were selected and compared. Additionally, Fig. 13 and Table 2 show that the model’s recognition accuracy for Annular, Churn, and Dispersed flow patterns is relatively low. The authors should provide an analysis of why these patterns are more challenging to identify.

Response: We appreciate your insightful observations regarding the convergence behavior of ED-DenseNet compared to the original DenseNet, as well as the classification challenges associated with Annular, Churn, and Dispersed flow patterns. We have addressed these concerns as follows:

1. Convergence Behavior of ED-DenseNet vs. Original DenseNet:

In Section 2.2, we have provided detailed training curves (refer to Fig. 13) that illustrate the convergence patterns of both ED-DenseNet and the original DenseNet models. The ED-DenseNet model achieves over 97% accuracy within the first 20 epochs, whereas the original DenseNet attains similar accuracy levels around the 40th to 50th epoch. This accelerated convergence of ED-DenseNet can be attributed to the integration of the Efficient Channel Attention (ECA) mechanism and dilated convolutions, which enhance feature extraction efficiency and model learning capacity.

2. Classification Challenges for Annular, Churn, and Dispersed Flow Patterns:

As highlighted in Fig. 14 and Table 3, the recognition accuracy for Annular, Churn, and Dispersed flow patterns is comparatively lower. These patterns present unique challenges due to their inherent characteristics:

Annular Flow: Characterized by a thin liquid film along the pipe walls with a central gas core, the subtle and dynamic nature of this film makes it difficult to capture in static images, leading to potential misclassification.

Churn Flow: Exhibits highly chaotic and oscillatory behavior with rapid transitions between liquid and gas phases, resulting in significant variability within this flow pattern.

Dispersed Flow: Consists of fine gas bubbles uniformly distributed within the liquid phase, making it challenging to distinguish from similar patterns like bubbly flow due to the small size and uniform distribution of the bubbles.

These intrinsic complexities contribute to the reduced classification accuracy for these specific flow patterns.

Comment 4: In Section 2.2 (Ablation Study), the authors mention the use of transfer learning as part of their methodology. While transfer learning is a common practice, its inclusion as a contribution is not sufficiently justified, especially since the ablation study does not explicitly evaluate its impact.

Response: Thank you for this valuable suggestion. We acknowledge that our initial description of transfer learning in Section 2.1(Original 2.2) lacked explicit evaluation and detailed justification. In response, we have made the following revisions:

1. Explicit Evaluation of Transfer Learning Impact: In the revised Section 2.1, we have added a comprehensive comparison between ED-DenseNet models trained with and without transfer learning. Specifically, we conducted an ablation experiment where: One ED-DenseNet model was initialized using pre-trained weights from DenseNet121 (transfer learning applied). Another ED-DenseNet model was trained entirely from scratch without pre-trained weights.

The results of this comparison are now included in Table 2. The key observations are: The ED-DenseNet model trained with transfer learning achieves an accuracy of 97.82%, whereas the model trained without transfer learning only reaches 94.89%, showing a significant 2.93% improvement. Training time decreased by approximately 93 minutes due to the faster convergence of the model with transfer learning.

2. Justification for Including Transfer Learning as a Contribution:

In the revised Introduction and the concluding paragraph of Section 2.2, we have emphasized: Given the relatively small size of the gas-liquid two-phase flow image dataset, transfer learning effectively mitigates the risk of overfitting and improves generalization. Transfer learning significantly accelerates convergence, which is critical for practical applications where training time and computational resources are limited. The explicit experimental evidence provided clearly demonstrates that transfer learning substantially enhances both recognition performance and training efficiency.

We sincerely thank the reviewers for their valuable feedback, which has significantly improved our manuscript. We believe these revisions address all concerns and improve the clarity, rigor, and completeness of our study. We look forward to your consideration of our revised submission.

Thank you again for your time and consideration.

Sincerely,

Jie Liu

---

## [Decision Letter · Decision Letter 1]

A flow pattern recognition method for gas-liquid two-phase flow based on dilated convolutional channel attention mechanism

PONE-D-24-47411R1

Dear Dr. liu,

We’re pleased to inform you that your manuscript has been judged scientifically suitable for publication and will be formally accepted for publication once it meets all outstanding technical requirements.

Kind regards,

Feng Ding

Academic Editor

PLOS ONE

Additional Editor Comments (optional):

Reviewers' comments:

Reviewer's Responses to Questions

**Comments to the Author**

Reviewer #1: All comments have been addressed

Reviewer #2: All comments have been addressed

2. Is the manuscript technically sound, and do the data support the conclusions?

Reviewer #1: Yes

Reviewer #2: Yes

3. Has the statistical analysis been performed appropriately and rigorously?

Reviewer #1: Yes

Reviewer #2: Yes

4. Have the authors made all data underlying the findings in their manuscript fully available?

Reviewer #1: Yes

Reviewer #2: Yes

5. Is the manuscript presented in an intelligible fashion and written in standard English?

Reviewer #1: Yes

Reviewer #2: Yes

Reviewer #1: (No Response)

Reviewer #2: (No Response)

**Do you want your identity to be public for this peer review?** For information about this choice, including consent withdrawal, please see our Privacy Policy

Reviewer #1: No

Reviewer #2: No

---

## [Editor Report · Acceptance letter]

PONE-D-24-47411R1

PLOS ONE

Dear Dr. Liu,

I'm pleased to inform you that your manuscript has been deemed suitable for publication in PLOS ONE. Congratulations! Your manuscript is now being handed over to our production team.

Kind regards,

on behalf of

Dr. Feng Ding

Academic Editor

PLOS ONE